# Human influenza A virus H1N1 in marine mammals in California, 2019

**Magdalena Plancarte[1‡], Ganna Kovalenko[2,3‡], Julie Baldassano[1], Ana L. Ramírez[1], Selina Carrillo[1], Pádraig J. Duignan[4], Ian Goodfellow[2], Eric Bortz[3], Jayeeta Dutta[5], Harm van Bakel[5,6], Lark L. Coffey[1]***

**1** Department of Pathology, Microbiology, and Immunology, School of Veterinary Medicine, University of California Davis, Davis, California, United States of America, **2** Division of Virology, Department of Pathology, University of Cambridge, Cambridge, United Kingdom, **3** Department of Biological Sciences, University of Alaska, Anchorage, Alaska, United States of America, **4** The Marine Mammal Center, Sausalito, California, United States of America, **5** Department of Genetics and Genomic Sciences, Icahn School of Medicine at Mount Sinai, New York, New York, United States of America, **6** Icahn Genomics Institute, Icahn School of Medicine at Mount Sinai, New York, New York, United States of America

‡ MP and GK share Co-first author on this work.
* lcoffey@ucdavis.edu

**Data Availability Statement:** Raw data showing serology results from individual marine mammals are located at https://zenodo.org/record/7697293#. ZAJIRy-B1UM. The GenBank accession numbers for the 8 IAV genome segments sequenced from a Northern elephant seal, IAV A/elephantseal/

## Abstract

From 2011–2018, we conducted surveillance in marine mammals along the California coast for influenza A virus (IAV), frequently detecting anti-influenza antibodies and intermittently detecting IAV. In spring 2019, this pattern changed. Despite no change in surveillance intensity, we detected IAV RNA in 10 samples in March and April, mostly in nasal and rectal swabs from northern elephant seals (*Mirounga angustirostris*). Although virus isolation was unsuccessful, IAV sequenced from one northern elephant seal nasal swab showed close genetic identity with pandemic H1N1 IAV subclade 6B.1A.1 that was concurrently circulating in humans in the 2018/19 influenza season. This represents the first report of human A (H1N1)pdm09 IAV in northern elephant seals since 2010, suggesting IAV continues to spill over from humans to pinnipeds.

## Introduction

Influenza A virus (IAV) infection has been reported in North American marine mammals since 1979 [1–4]. More than 300,000 marine mammals, including California sea lions (CSL), Steller sea lions, Northern and Guadalupe fur seals, Pacific harbor seals (PHS), and northern elephant seals (NES), live in the eastern North Pacific Ocean and molt and birth along mainland and island coastlines of California [5]. Pinnipeds in California are in contact with influenza virus reservoirs, including avian and mammalian hosts [6]. In 2009, pandemic H1N1 IAV emerged in the United States, spread worldwide, and was detected in NES in California in 2010 [6] and in northern sea otters in Washington state in 2011 [7]. Through our IAV surveillance in marine mammals on the California coast from 2011–2018, we rarely detected IAV RNA that indicates current or recent infection. IAV antibody detection, which reveals previous IAV infection, is more common. This study focuses on a cluster of IAV RNA positive

California/ES4506NS/2019(H1N1) are MW132314 and MW132331-132337.

**Funding:** This project was funded by the National Institutes of Health Centers of Excellence for Influenza Research and Surveillance (CEIRS) HHSN272201400008C, HHSN266200700010C, and HHSN272201400006C. Additional support was provided by the NIGMS Alaska INBRE program P20GM103395. GK and IG are supported by grants from the Wellcome Trust (Refs: 207498/Z/17/Z and 206298/B/17/Z). Financial support for JB was provided by the Students Training in Advanced Research (STAR) Program through a UC Davis School of Veterinary Medicine Endowment Fund. The funders had no role in study design, data collection and analysis, decision to publish, or preparation of the manuscript.

**Competing interests:** The authors have declared that no competing interests exist.

detections in two species of marine mammals stranded on California coasts in spring 2019. Sequence analyses of IAV from one NES show close identity with 2018 and 2019 human H1N1 IAV.

## Materials and methods

### Sample collections

Marine mammals were sampled as part of ongoing IAV surveillance from 2011–2019. Nasal and rectal swabs and blood, were collected at intake from pinnipeds who became stranded along the California coast and were admitted for rehabilitation to The Marine Mammal Center (TMMC) in Sausalito, CA. Marine mammal sampling was performed by TMMC staff with prior authorization from the National Oceanic and Atmospheric Administration National Marine Fisheries Service Marine Mammal Protection Act. In spring 2019, after detecting IAV RNA in several samples, we modified sampling to add a second blood sample collection from some animals 2 to 3 weeks after collecting the first sample. Nasal and rectal swabs were placed individually or combined in vials containing 1.5 mL of viral transport media (VTM). Blood was collected in serum separator tubes which were centrifuged at 4000 revolutions per minute (rpm) for 10 minutes (min), after which serum was transferred to a cryovial. Samples were refrigerated for up to 1 week before being transferred to a laboratory, where they were immediately processed or stored at −80˚C.

### Serology assays

Sera diluted 1:10 were screened in a single replicate for IAV antibody directed against a conserved epitope of the IAV nucleoprotein (NP) using an enzyme-linked immunosorbent assay (ELISA) kit ID-Screen® Influenza A Antibody Competition Multi-species kit, (IDvet, Grabels, France) following the manufacturer's instructions. Positive and negative controls provided with the kit were included on each plate. An ELX808 BioTek Spectrophotometer (BioTek Instruments, Winooski, VT) was used to measure absorbance. A sample was reported to contain IAV NP antibody when the absorbance ratio of the test **s**ample to the **n**egative control (**S/N** ratio) was less than 0.45, as previously established [8]. For a subset of animals in 2019 with detectable ELISA IAV NP in serum at intake or in the second sample, hemagglutination inhibition (HI) assays were also performed to identify the IAV subtype per established criteria [9]. Each sample was tested in triplicate by HI with each of 3 IAV strains: an H1N1 Northern elephant seal isolate from 2010 (A/Elephant seal/California/1/2010(H1N1)), an H3N8 harbor seal isolate from 2011 (A/harbor seal/New Hampshire/179629/2011(H3N8)), and an H5N2 mallard duck isolate from 2010 (A/mallard/California/2396/2010(H5N2)) following an established protocol [10]. The H1N1 and H3N8 subtypes were selected since they were previously reported in marine mammals, and the H5N2 subtype was used as it is common in water birds that share shoreline environments with pinnipeds. Positive control sera from a ferret that had been experimentally inoculated with a pandemic H1N1 strain (A/California/04/2009) was provided by Dr. Randy Albrecht, Icahn School of Medicine at Mount Sinai, New York. For HI assays serum was diluted 1:4 with receptor destroying enzyme (RDE, Denka Seiken, Tokyo) and incubated for 18 hours at 37˚C, followed by 30 min at 56˚C. Sera were then serially diluted 2-fold in phosphate-buffered saline (PBS) and incubated with 4 hemagglutination units of virus for 60 min at room temperature. Chicken erythrocytes (Lampire, Pipersville) at a final concentration of 0.25% in PBS were added to sera and virus, and the mixture was incubated for 60 minutes at 27˚C. Treated sera were also tested for hemagglutination in the absence of virus to verify effective RDE treatment and to ensure the absence of nonspecific

hemagglutination. Serum was defined as negative for HI when a titer below 1:8 was detected. Non-reactive samples are reported as <8.

### IAV RNA detection

RNA was extracted using a MagMAX-96 AI/ND Viral RNA Isolation Kit (Applied Biosystems, Foster City, CA) using a KingFisher Magnetic Particle Processor (Thermo Scientific, Waltham, MA). RNA extracts were screened for IAV using the AgPath-IDTM One Step RT-PCR mix (Applied Biosystems, Foster City, CA) and an ABI 7500 real-time PCR System (Applied Biosystems, Foster City, CA). The real-time reverse transcription polymerase chain reaction (RRT-PCR) targets a conserved region of the IAV matrix gene [11]. Each RNA extract was tested in one replicate. Each RRT-PCR plate included an IAV isolate from cell culture as a positive control and VTM diluent as a negative control. Samples with a cycle threshold (Ct) value <45 were considered positive.

### Virus isolation

Virus isolation was attempted for all RRT-PCR positive samples by inoculation into embryonated chicken eggs (Charles River, CT, USA) and Madin-Darby canine kidney cells (MDCK, ATCC, Manassas, Virginia). Each sample was inoculated into eggs and cells twice using a described protocol [6]. After each isolation attempt, egg samples were screened by RRT-PCR for IAV RNA, and cells were observed for cytopathic effects characteristic of IAV infection.

### Hemagglutinin (HA) subtyping and genome sequencing

We amplified and sequenced a portion of the IAV HA gene from RRT-PCR positive samples before conducting whole-genome sequencing that requires more time and effort. HA subtyping was performed using a protocol modified from [12] that generates a 640 base pair (bp) PCR product. RNA was used to make first-strand cDNA using random primers and M-MLV reverse transcriptase kit (Invitrogen,Carlsbad, CA). First, 2.0 μl of 0.5μl/μl random primers, 1.0μl 10mM dNTP, 4.0 μl RNA and DEPC-water were mixed to a total 13 μl volume. The mixture was incubated for 10 min at 65˚C and placed on ice. Next, 4.0 μl of this reaction was added to 4.0 μl of 5x buffer, 2.0μl of 0.1 M DTT, 1μl RNAse inhibitor and 1.0 M-MLV reverse transcriptase to a total volume of 21 μl. The reaction was incubated at 25˚C for 10 min, 37˚C for 50 min and 65˚C for 10 min. For PCR, the forward primer 5′ `GGRATGRTHGAYGGNTGGTAYGG` 3′ was modified from a validated primer [12] to include HA sequences detected in California. The HARK reverse primer 5′ `ATATGGCGCCGTATTA GTAGAAACAAGGGTGTTTT` 3′ was first reported in Bragstad *et al.* [13]. These primers are designed to detect all HA subtypes. The 25 μL PCR reaction used 0.08 U Amplitaq Gold polymerase (Invitrogen, Carlsbad CA) and 10x buffer without MgCl$_2$, 1.5 mM MgCl$_2$, 0.2 mM dNTP mix, 0.6 μM of each primer, and 7 μL of RNA extract from each sample. The PCR conditions were 10 min at 95˚C, followed by 45 cycles of 94˚C for 1 min, 58˚C for 1 min, and 72˚C for 1 min, with a final extension of 72˚C for 7 min. Following electrophoresis on a 1.5% agarose gel, the 640 bp band was excised and purified using a QIAquick gel extraction kit (Qiagen, Valencia, CA) with an elution volume of 30 μL. Direct Sanger sequencing of the amplicons was performed using the HARK reverse primer. Sequences were BLASTed (http://www. ncbi.nlm.nih.gov/BLAST) to identify the most similar HA sequences in GenBank. Whole-genome sequencing was next performed using next generation sequencing from samples where HA sequencing was successful using methods described previously [14]. Briefly, RNA was extracted using the QIAamp Viral RNA Mini Kit (Qiagen, Valencia, CA). Following extraction, MS-RT-PCR amplification of influenza viral genome segments was performed with

the Superscript III high-fidelity RT-PCR kit (Invitrogen, Carlsbad CA) according to manufacturer's instructions using the Opti1 primer set: Opti1-F1 5′ GTTACGCGCCAGCAAAAGCA GG, Opti1-F2 5′ GTTACGCGCCAGCGAAAGCAGG, Opti1-R1 5′ GTTACGCGCCAGTAGAAA CAAGG. These are influenza-specific universal primers that are complementary to the conserved 12–13 nucleotides at the end of all 8 genomic segments. The RT-PCR amplification parameters were: 2 min at 55˚C, 60 min at 42˚C, and 2 min at 94˚C, followed by 5 cycles of (94˚C/30 s; 44˚C/30 s; 68˚C/3.5 min), 26 cycles of (94˚C/30 s; 57˚C/30 s; 68˚C/3.5 min), and a final extension for 10 min at 68˚C. Amplicons were purified with 0.45X volume of Agencourt AMPure XP beads (Beckman Coulter, Brea, CA) and visualized on the Agilent Bioanalyzer or a 2% agarose gel. The concentration of purified amplicons was measured using the Qubit High Sensitivity dsDNA kit. After samples were normalized to a concentration of 0.2 ng/μl, adapters were added by tagmentation using the Nextera XT DNA library preparation kit (Illumina). Samples were purified using 0.7X of Agencourt AMPure XP Magnetic Beads and fragment size distributions were analyzed on a Bioanalyzer using the High Sensitivity DNA kit (Agilent). After bead-based normalization (Illumina, Hayward, CA) according to the manufacturer's protocols, sequence-ready libraries were sequenced in a paired-end run using the MiSeq v2, 300cycle reagent kit (Illumina). Complete genomes were assembled from next generation Illumina reads using a custom influenza genome assembly pipeline [14]. The genome assembly code is available at https://bitbucket.org/bakellab/flugap.

## Phylogenetic analyses

Phylogenetic analyses for all eight IAV segments were performed. We downloaded all available human A(H1N1)pdm09 sequences sampled between January 2016 and December 2021 from the EpiFlu database hosted by the Global Initiative on Sharing All Influenza Data (GISAID; platform.gisaid.org) that were available on July 1, 2022. We removed all sequences missing an exact sampling date, duplicates (by strain names) and those with non-completed genome segments. Sequences were grouped according to each geographic location (South America, North America, Europe, Africa, Asia and Oceania) and by year (2016, 2017, 2018, 2019, 2020, 2021). CD-HIT v4.8.1 was then used to remove highly similar sequences (sharing 99–100% nucleotide identity) [15]. This allowed us to downsample the datasets and select representative sequences from each group to analyze genetic diversity. Additionally, two available H1N1pdm09 sequences from NES in 2010 were included in the analyses along with a reference sequence set for the clade/subclade assignation. The reference dataset was prepared using a nomenclature method introduced by the European Centre for Disease Prevention and Control (ECDC) that classifies a genetic clade/subclade based on unique amino acid substitutions in the HA1 or HA2 proteins (**S1 Table**) [16]. Sequence alignments were constructed using MAFFT v7.490 [17]. The alignments were trimmed and only gene coding regions were used. Phylogenetic trees for each segment were constructed by the maximum likelihood (ML) method using IQ-TREE with substitution model selection (ModelFinder implemented in IQ-TREE) option and 1000 bootstraps [18].

Given that our ML trees all showed that the IAV genome detected in the NES in this study is not a reassortant virus and belongs to A/H1N1 pandemic influenza type, we performed concatenation of all eight segments. The evolutionary relationships and timescale of the concatenated eight segments were inferred using a Bayesian Markov chain Monte Carlo (MCMC) method, as implemented in the BEAST v1.10.4 package [19]. The final dataset of 165 sequences were downsampled with the CD-HIT by geographic origin and year of sampling along with two available A(H1N1)pdm09 sequences sampled from NES in 2010 and the reference set. After the MAFFT aligning, we trimmed sequences to 13,154 bp. A strict molecular

clock was used, under the general time reversible model allowing for rate heterogeneity among sites and proportion of invariable sites (GTR+G+I). The MCMC was run for 100 million iterations, with subsampling every 10,000 iterations at least two times. All parameters reached convergence, as assessed visually using Tracer v1.7.1 [20], with statistical uncertainty reflected by values of the 95% Highest Posterior Density (HPD) interval. The initial 10% of the chain was removed as burn-in, and maximum clade credibility (MCC) tree was summarized using TreeAnnotator v1.10.4. The tree was visualized and annotated using FigTree v1.4.4 (http://tree.bio.ed.ac.uk/software/figtree). Additionally, to assess the reliability of the phylogenetic inference of the concatenated segments, an additional Bayesian tree was estimated for the H1 segment separately with the same dataset that was used for the concatenation approach and with the same model parameters.

### Data reporting and statistical analyses

Raw data showing serology results from individual marine mammals are located at https://zenodo.org/record/7697293#.ZAJlRy-B1UM. The GenBank accession numbers for the 8 IAV genome segments sequenced from a Northern elephant seal, IAV A/elephantseal/California/ES4506NS/2019(H1N1) are MW132314 and MW132331-132337. Rates of ELISA seropositivity were compared using Chi-squared statistics with Yates' correction. P values <0.05 were considered statistically significant. Statistical analyses were performed using GraphPad Prism.

## Results

After the first detection of pandemic H1N1 IAV in two NES in 2010, we performed continuous IAV surveillance in multiple pinniped species stranded on the California coast and admitted to TMMC from 2011–2019. Our approach entails screening marine mammals for IAV antibody in serum via ELISA and sometimes HI as well as viral RNA in nasal and rectal swabs after RNA isolation and RRT-PCR. We tested at least 800 swabs samples annually. Most samples were from CSL (*Zalophus californianus*), NES (*Mirounga angustirostris*), PHS (*Phoca vitulina)*, and Northern fur seals (*Callorhinus ursinus*). From 2011–2018, we detected 13 IAV RNA-positive nasal and/or rectal swabs (0 in 2011, 1 in 2012, 0 in 2013–2015, 5 in 2016, 3 in 2017, and 4 in 2018). In 2019, we detected 10 IAV RNA-positive nasal and/or rectal swabs, 9 of which were from animals admitted during a 6-week period between March 12 and April 27, 2019 (**Table 1**; 2016–2018 data included for historical reference). Given the increased number of IAV RNA detections in 2018–2019 compared to 2011–2018 and the temporal clustering of IAV RNA detections in spring 2019, this manuscript focuses on data from animals stranded during the 2018–2019 period.

   We detected IAV NP antibody by ELISA in 42% (2018) and 54% (2019) of NES, in 6% (2018) and 0% (2019) of PHS, and in 6% (2018) and 21% (2019) of CSL (**Fig 1A**). There was no difference in the overall rate of ELISA seropositivity between 2018 and 2019 in NES or PHS (p>0.05, Chi-squared). The rate of ELISA seropositivity in CSL was significantly higher in 2019 compared to 2018 (p = 0.04, Chi-squared). Pups and weanlings represented most stranded animals and had higher rates of IAV NP antibody compared to juveniles, yearlings, and adults. HI was performed on IAV ELISA-reactive sera from 70 animals (**Table 2**). Each serum sample was tested by HI using 3 IAV subtypes: H1N1 and H3N8, selected since they were previously reported in marine mammals, and H5N2 which is common in water birds that share shore environments with marine mammals. Of the 70 marine mammals for which HI titers were determined, 60 had H1N1 HI antibody titers at least 4-fold higher than H3N8 and H5N2 titers. Seven were H1N1 reactive above the limit of detection of 8 but did not have titers that were 4-fold higher than the other two subtypes. Two samples had detectable H3N8

**Table 1. Metadata for IAV RNA positive marine mammals on the California coast, 2016–2019.** The second sample on which ELISA or HI was performed upon was collected between 2 to 3 weeks after the admit sample. CSL is California sea lion, NES is northern elephant seal, NFS is northern fur seal, PHS is Pacific harbor seal. F is female, M is male, N/A indicates not applicable, Neg is negative, Pos is positive, HI is hemagglutination inhibition, ELISA is enzyme linked immunosorbent assay, RRT-PCR is real time reverse transcription polymerase chain reaction, Ct is cycle threshold. Asterisks indicate that HI was performed on the first sample since for some animals a second sample was not available. The number after each H designation indicates the hemagglutinin subtype used in the HI assay. A non-HI reactive sample is annotated at <8, where a 1:8 dilution was the lowest tested. Each RRT-PCR Ct and ELISA represent testing of a single replicate for each sample. For HI, each sample was tested in triplicate and the mean HI titer is shown.

| Animal ID | IAV RRT-PCR Ct | ELISA | | Mean HI titer for second sample | | | Age Class | Sex | Species | Strand County | Strand City | Admit Date | Disposition | | Diagnosis | Cause of Death |
|---|---|---|---|---|---|---|---|---|---|---|---|---|---|---|---|---|
| | | admit | second | H1 | H3 | H5 | | | | | | | Date | Status | | |
| CSL-13160 | 37.6 | Neg | ND | ND | ND | ND | Subadult | F | CSL | San Mateo | Pescadero | 06/05/2016 | 06/18/2016 | Died in Treatment | malnutrition | malnutrition |
| CSL-13161 | 42.0 | Neg | ND | ND | ND | ND | Adult | F | CSL | Santa Cruz | Watsonville | 06/06/2016 | 06/10/2016 | Euthanasia | N/A | euthanasia, domoic acid toxicity |
| CSL-13198 | 37.7 | Neg | ND | ND | ND | ND | Yearling | F | CSL | Monterey | Moss Landing | 07/04/2016 | 07/05/2016 | Euthanasia | N/A | gunshot |
| CSL-13203 | 36.8 | NA | ND | ND | ND | ND | Adult | F | CSL | San Luis Obispo | Morro Bay | 07/12/2016 | 07/31/2016 | Euthanasia | N/A | domoic acid toxicity |
| CSL-13207 | 37.1 | NA | ND | ND | ND | ND | Adult | F | CSL | Monterey | Pacific Grove | 07/14/2016 | 07/16/2016 | Died in Treatment | N/A | renal failure, neoplasia |
| ES-4068 | 36.5 | Neg | ND | ND | ND | ND | Pup | M | NES | Santa Cruz | Davenport | 01/24/2017 | 06/07/2017 | Released | malnutrition, maternal separation | N/A |
| ES-4121 | 35.8 | Neg | ND | ND | ND | ND | Pup | M | NES | Monterey | Pacific Grove | 03/20/2017 | 04/27/2017 | Released | malnutrition | N/A |
| NFS-435 | 35.7 | Neg | ND | ND | ND | ND | Pup | F | NFS | Monterey | Monterey | 12/21/2017 | 02/02/2018 | Released | malnutrition | N/A |
| ES-4254 | 35.4 | Neg | ND | ND | ND | ND | Pup | F | NES | Santa Cruz | Davenport | 01/22/2018 | 06/22/2018 | Released | maternal, separation, malnutrition, | N/A |
| HS-2754 | 35.3 | Neg | ND | ND | ND | ND | Pup | M | PHS | Marin | Bolinas | 02/14/2018 | 02/14/2018 | Died in Treatment | N/A | prematurity, maternal separation |
| ES-4311 | 36.5 | Pos | ND | 32* | 8* | <8* | Pup | M | NES | Monterey | Monterey | 03/26/2018 | 06/22/2018 | Released | malnutrition | N/A |
| CSL-14130 | 37.6 | Neg | ND | ND | ND | ND | Pup | M | CSL | San Mateo | Princeton-by-the-Sea | 12/17/2018 | 01/22/2019 | Released | malnutrition, maternal separation | N/A |
| ES-4424 | 35.8 | Pos | ND | ND | ND | ND | Pup | F | NES | San Luis Obispo | San Simeon | 3/12/2019 | 4/24/2019 | Released | malnutrition | N/A |
| ES-4506 | 20.9 | Neg | ND | ND | ND | ND | Pup | M | NES | San Luis Obispo | Avila Beach | 4/9/2019 | 7/17/2019 | Released | malnutrition, oil | N/A |

*(Continued)*

Table 1. (Continued)

| Animal ID | IAV RRT-PCR Ct | ELISA | | Mean HI titer for second sample | | | Age Class | Sex | Species | Strand County | Strand City | Admit Date | Disposition | | Diagnosis | Cause of Death |
|---|---|---|---|---|---|---|---|---|---|---|---|---|---|---|---|---|
| | | admit | second | H1 | H3 | H5 | | | | | | | Date | Status | | |
| ES-4507 | 28.6 | Pos | ND | 8* | <8* | <8* | Pup | M | NES | Santa Cruz | Live Oak | 4/9/ 2019 | 4/26/ 2019 | Euthanized | malnutrition, otostrongyliasis | otostrongyliasis |
| ES-4509 | 32.4 | Neg | Pos | 64 | 11 | <8 | Pup | F | NES | San Mateo | Pacifica | 4/10/ 2019 | 5/21/ 2019 | Released | malnutrition | N/A |
| ES-4523 | 34.7 | Pos | Pos | 128 | <8 | <8 | Pup | F | NES | Monterey | Pacific Grove | 4/14/ 2019 | 7/17/ 2019 | Released | malnutrition | N/A |
| ES-4527 | 32.6 | Neg | Pos | 256 | <8 | <8 | Pup | F | NES | Sonoma | Fort Ross | 4/15/ 2019 | 6/12/ 2019 | Released | malnutrition, otostrongyliasis, abscess | N/A |
| ES-4530 | 32.1 | Neg | Neg | ND | ND | ND | Pup | M | NES | San Mateo | Montara | 4/16/ 2019 | 6/1/ 2019 | Released | malnutrition, otostrongyliasis | N/A |
| ES-4538 | 28.2 | Pos | ND | 128* | <8 | <8 | Pup | F | NES | San Luis Obispo | San Simeon | 4/20/ 2019 | 6/12/ 2019 | Released | malnutrition, trauma, unknown | N/A |
| ES-4539 | 24.2 | Neg | Neg | >512 | <8 | <8 | Pup | F | NES | San Luis Obispo | Cayucos | 4/20/ 2019 | 5/2/ 2019 | Euthanized | malnutrition, trauma, unknown | congenital defect |
| HS-2859 | 32.0 | Neg | ND | ND | ND | ND | Pup | F | PHS | San Mateo | Pacifica | 4/27/ 2019 | 6/18/ 2019 | Released | maternal separation, malnutrition | N/A |

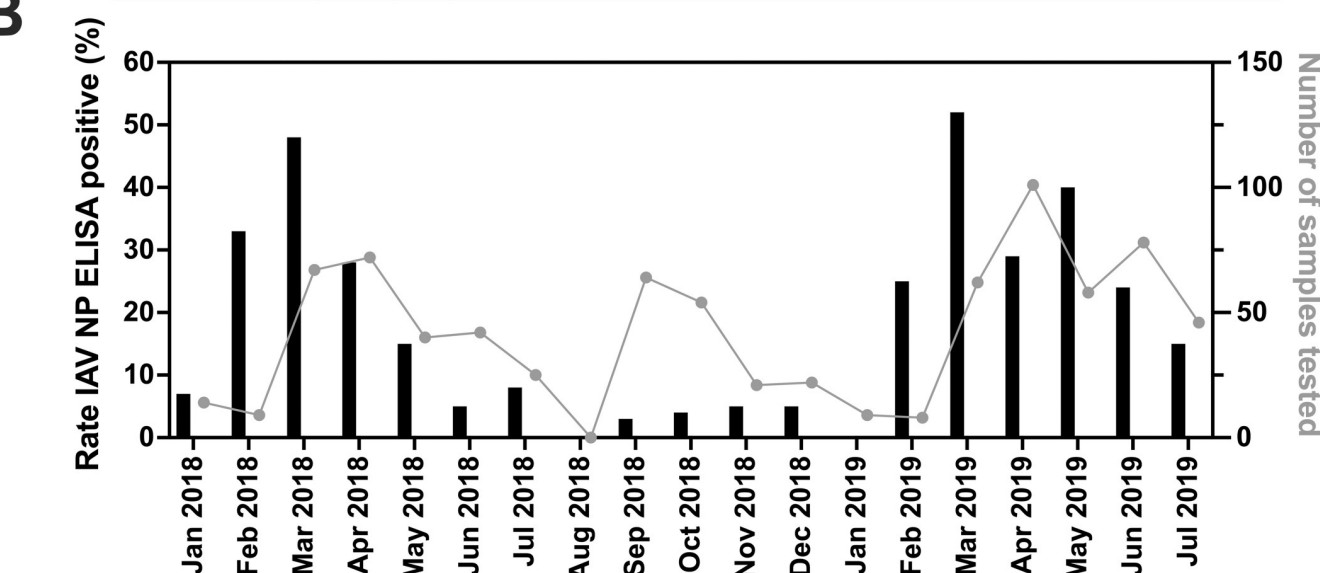

| Species | | Northern elephant seal | Pacific harbor seal | California sea lion |
|---|---|---|---|---|
| Year | Assay | \% animals with detectable antibody (number positive/total) | | |
| 2018 | ELISA | 42 (56/135) | 0 (0/29) | 6 (17/267) |
| 2019 | | 54 (74/137) | 6 (3/55) | 21 (36/170) |
| | Chi-square | p>0.05 | p>0.05 | p=0.04 |
| | Age class | | | |
| 2018 and 2019 | Weaner and pup | 49 (130/264) | 2(2/83) | 16 (16/99) |
| | Juvenile, yearling, and adult | 0 (0/8) | 100 (1/1) | 11 (37/338) |

| Species | | Northern elephant seal | Pacific harbor seal | California sea lion |
|---|---|---|---|---|
| Year | Assay | \% animals with detectable antibody (number positive/total) | | |
| 2018 | HI | 85 (25/29) | nd | nd |
| 2019 | | 93 (34/37) | 50 (1/2) | nd |

**Fig 1. Influenza A virus antibody detections in marine mammals, California, 2018–2019. A**) Rates of IAV nucleoprotein (NP) enzyme-linked immunosorbent assay (ELISA) and H1N1 hemagglutination inhibition (HI) reactivity in marine mammal sera from January 2018 to July 2019. ELISA antibody detections by age class of marine mammals are also shown. nd indicates not done. **B)** IAV NP ELISA seroprevalence by month for the study period.

titers that were less than 4-fold higher than H1N1 titers, and one sample was not reactive with any IAV subtype by HI. Most (85% in 2018, 93% in 2019) ELISA-reactive NES sera also contained detectable H1N1 HI titers. The IAV NP ELISA positivity rate was highest in the winter months, exceeding 20% from February to April in 2018 and from February to June in 2019, and peaking near 50% in March of both years (**Fig 1B**). This seasonal bias in seropositivity may reflect higher sampling in winter, where NES are pelagic the rest of the year and therefore not available for sampling. Together these data indicate that H1N1 IAV infection is common in multiple pinniped species as evidenced by infection- or maternally-derived antibodies detected in pups and weanlings.

In 2018, we assayed 1385 nasal and rectal swabs from 969 CSL, 332 NES, and 84 PHS; four contained detectable IAV RNA. In 2019, we assayed 822 nasal and rectal swabs; 539 from CSL, 201 from NES, and 82 from PHS. From the 2019 samples, we detected IAV RNA in the nasal swab from one Northern elephant seal, combined nasal and rectal swabs from additional eight NES, and one nasal swab from a PHS, representing 10/822 (1.2%) of the 2019 total from all three species. In 2019, The IAV RNA-positive animals all stranded in March and April. All

**Table 2. IAV antibody hemagglutination inhibition (HI) assays detections in ELISA-IAV reactive Northern elephant seals on California coasts in 2018–2019.** Animals shaded in grey also contained IAV RNA detectable by RRT-PCR; the same HI data for those animals is also reproduced in Table 2 for comparison with RNA values. Ferret serum from an animal that was experimentally inoculated with IAV H1N1 was used as a positive control. Positive control sera for H3N8 and H5N2 were not available. The negative control consisted of serum diluent.

| Animal ID | Mean HI titer | | | Animal ID | Mean HI titer | | |
|---|---|---|---|---|---|---|---|
| | H1N1 | H3N8 | H5N2 | | H1N1 | H3N8 | H5N2 |
| ES-4266 | 256 | <8 | <8 | ES-4505 | 512 | <8 | <8 |
| ES-4280 | 352 | 11 | 8 | ES-4507 | 8 | <8 | <8 |
| ES-4284 | >512 | <8 | <8 | ES-4509 | 48 | 11 | <8 |
| ES-4298 | 64 | 16 | <8 | ES-4510 | 8 | <8 | <8 |
| ES-4302 | 128 | 32 | 32 | ES-4511 | 256 | 21 | <8 |
| ES-4307 | 8 | 16 | 8 | ES-4513 | 48 | <8 | <8 |
| ES-4310 | 16 | <8 | <8 | ES-4514 | 256 | <8 | <8 |
| ES-4311 | 32 | 8 | <8 | ES-4516 | 256 | <8 | <8 |
| ES-4314 | 128 | <8 | <8 | ES-4518 | 512 | <8 | <8 |
| ES-4318 | 128 | 8 | <8 | ES-4520 | 256 | <8 | <8 |
| ES-4319 | 64 | 8 | <8 | ES-4521 | 256 | <8 | <8 |
| ES-4324 | 128 | <8 | <8 | ES-4523 | 256 | <8 | <8 |
| ES-4325 | 0 | 8 | <8 | ES-4524 | 128 | <8 | <8 |
| ES-4327 | 64 | <8 | <8 | ES-4526 | 512 | <8 | <8 |
| ES-4330 | 16 | <8 | <8 | ES-4527 | 256 | <8 | <8 |
| ES-4333 | 128 | <8 | <8 | ES-4529 | 128 | <8 | <8 |
| ES-4335 | 16 | <8 | <8 | ES-4531 | 128 | <8 | <8 |
| ES-4339 | 32 | 32 | <8 | ES-4532 | 8 | <8 | <8 |
| ES-4342 | 16 | 8 | <8 | ES-4535 | 256 | <8 | <8 |
| ES-4343 | 128 | <8 | <8 | ES-4537 | 128 | <8 | <8 |
| ES-4347 | 128 | <8 | <8 | ES-4538 | 128 | <8 | <8 |
| ES-4348 | 128 | 32 | 32 | ES-4539 | >512 | <8 | <8 |
| ES-4349 | 4 | <8 | <8 | ES-4542 | 64 | <8 | <8 |
| ES-4354 | 256 | <8 | <8 | ES-4543 | 512 | 8 | <8 |
| ES-4357 | <8 | <8 | <8 | ES-4544 | 256 | <8 | <8 |
| ES-4373 | 512 | 32 | 0 | ES-4545 | 512 | <8 | <8 |
| ES-4383 | 427 | 48 | 16 | ES-4550 | 128 | <8 | <8 |
| ES-4387 | 64 | <8 | <8 | ES-4552 | 64 | <8 | <8 |
| ES-4389 | 512 | <8 | <8 | ES-4558 | 512 | 11 | <8 |
| ES-4470 | 128 | <8 | <8 | ES-4562 | 320 | 21 | 16 |
| ES-4472 | 128 | <8 | <8 | ES-4564 | 256 | 8 | 8 |
| ES-4477 | 384 | <8 | <8 | HS-2861 | <8 | 8 | <8 |
| ES-4479 | 512 | <8 | <8 | HS-2883 | >512 | 32 | 32 |
| ES-4482 | 256 | <8 | <8 | H1N1 positive control | >512 | <8 | <8 |
| ES-4490 | 256 | <8 | <8 | Negative control | <8 | <8 | <8 |

animals were pups, and six of the ten animals were female. All showed signs of malnutrition and most were released after rehabilitation. For the two animals that were euthanized, the cause of death was not IAV-related (**Table 1**). Each sample yielded a positive IAV matrix gene RRT-PCR result, with Ct values ranging from 20.9 to 35.8. The stranding locations of the 10 IAV RRT-PCR positive marine mammals from 2019 spanned the California coast (**Fig 2**). Unfortunately, we were unsuccessful isolating infectious IAV from any RRT-PCR positive swab samples after inoculation into embryonated chicken eggs and MDCK cells. We amplified

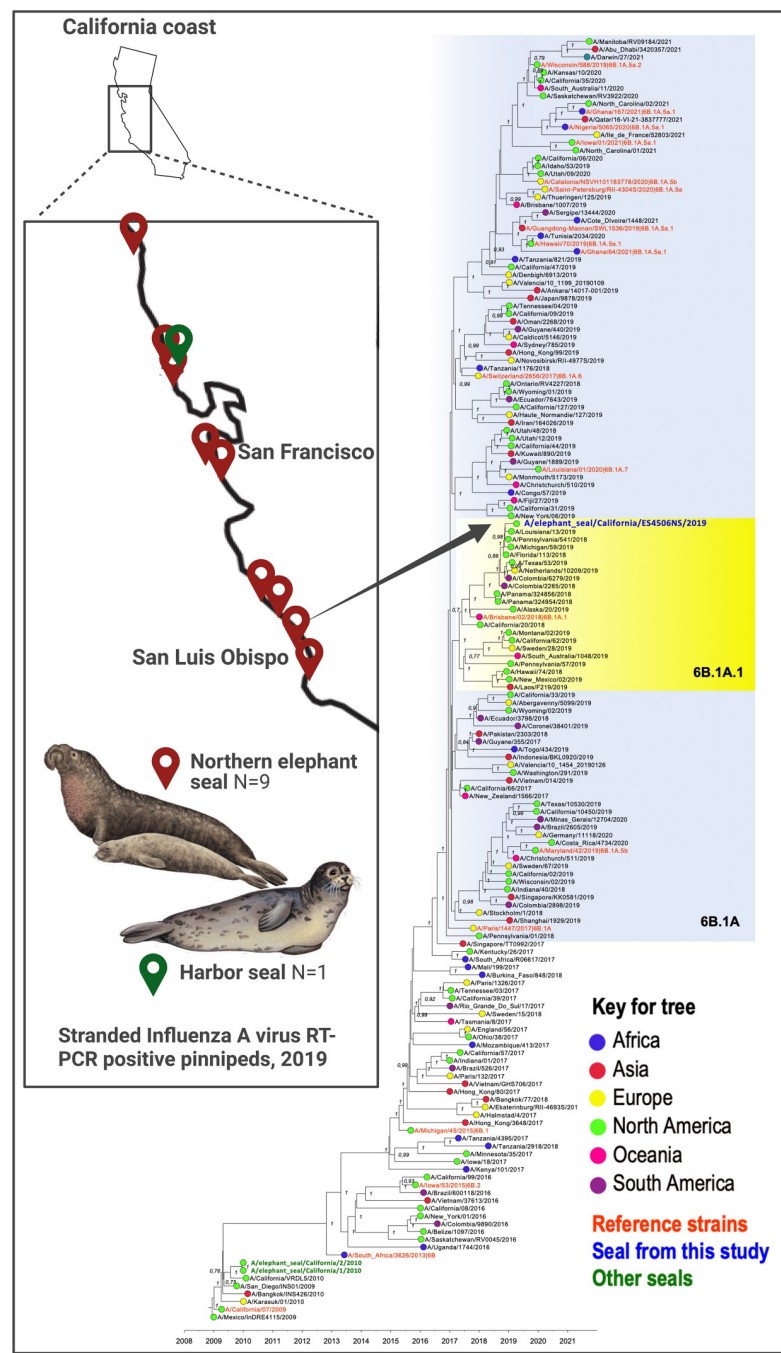

**Fig 2. Influenza A virus RNA detections in marine mammals, California, 2019.** Map shows locations of stranded marine mammals on California coasts that contained IAV RNA in nasal swabs. The tree is a time-scaled Maximum Clade Credibility phylogeny representing concatenated coding regions of all 8 A(H1N1)pdm09 IAV segments. A total of 165 representative pandemic H1N1 isolates sampled globally are included in the tree. Sequences in black text are from humans. The sequence in blue text shows the position of the IAV genome from a Northern elephant seal in this study. Green text shows sequences of IAV from other seals not part of this study that were sampled in 2010. The yellow shading shows clade 6B.1A/subclade 6B.1A.1 IAV, represented by the reference 2019–2020 vaccine virus A/Brisbane/02/2018. Bayesian posterior probability values >70% are included for key nodes. The tree is rooted through the assumption of a strict molecular clock, such that tip times represent the time (year/month/day) of sampling. The source of the Northern elephant seal illustration was https://www.sciencedirect.com/topics/agricultural-and-biological-sciences/northern-elephant-seal. The figure was generated using Biorender.

and sequenced a 640 bp portion of the IAV HA gene from the nasal swab from one NES (ES-4506NS) that exhibited the lowest RRT-PCR Ct value (**Table 1**). The complete IAV genome was then obtained by sequencing multi-segment RT-PCR reactions generated from the nasal swab sample. This sequence was named A/Northern elephant seal/California/ES4506NS/2019 (H1N1). The serum sample from that animal at admission to TMMC was ELISA IAV NP antibody negative. Unfortunately, the second sample was not collected to evaluate whether seroconversion occurred during rehabilitation.

The IAV genome detected in the NES in this study is closely related to pandemic A(H1N1) pdm09 virus that circulated in humans during the 2018–19 influenza season and is derived from clade 6B.1A/subclade 6B.1A1. A Maximum Clade Credibility phylogeny constructed from the concatenated coding regions of all eight segments (**Fig 2**) shows that the closest related strains are from the United States (e.g. A/Pennsylvania/541/2018 and A/Michigan/59/2019) with highest nucleotide identity of 99.5% across the whole genome and supported by >90% Bayesian posterior probabilities. The A/Northern elephant seal/California/ES4506NS/2019(H1N1) from the NES in this study belongs to clade 6B.1A/subclade 6B.1A.1, a sub-group of H1N1pdm09 viruses that began circulating in the 2018/19 influenza season and which are characterized by a S183P mutation in the hemagglutinin (HA) protein. This subclade was contemporaneously circulating in the Northern Hemisphere in 2018/19 influenza season and is represented by the reference strain 2019–2020 vaccine virus A/Brisbane/02/2018. Separate analyses of each of the eight segments (PB2, PB1, PA, H1, NP, N1, M, and NS) of the H1N1 virus from the NES in this study was inferred using the H1 classified phylogeny as a reference and shows that all gene segments group within the same clade (**S1-S8 Figs in S1 File**). In particular, the H1 HA phylogeny maintains similar topological accuracy to that of the concatenated tree, where individual clades are supported by high Bayesian posterior probabilities (BPP > 90%) (**S9 Fig in S2 File**). No reassortant events were detected in the H1N1pdm09 NES virus in our analyses. The predicted HA amino acid sequence of the NES virus described here is identical to the consensus of concurrently circulating human strains in the 6B.1A.1 subclade. To predict the timeframe of a possible spillover event(s) into the NES, the time of the most recent common ancestor (tMRCA) was estimated for both the concatenated coding region sequence and the HA alone. For the concatenated H1N1pdm09 genome, the tMRCA between the NES IAV and the closest human isolate A/Pennsylvania/541/2018 IAV was estimated to be November 19, 2018 (2018.8695 decimal year) with a 95% Highest Posterior Density (HPD) October 20, 2018 –December 6, 2018 (2018.8047–2018.9332). The tMRCA for the HA alone was in the same timeframe and comparably estimated to be November 11, 2018, ranging between the end of September and late December 2018 (2018.8642 decimal year (95% HPD 2018.7402–2018.9814) (**S9 Fig in S2 File**).

## Discussion

This study represents the second detection of pandemic H1N1pdm09 IAV RNA in marine mammals in California. Like the 2010 detections in two NES, the locations and exact timing of exposures are not known. In this study, samples from other species, like gulls or shore birds sharing near or offshore environments with pinnipeds, were not available for IAV testing.

Consistent with 2010, where two IAV RNA-positive NES were detected in two consecutive weeks in late April and early May [6], all the IAV RNA-positive animals in this study were detected in spring (March and April 2019), suggesting that H1N1pdm09 IAV circulation in these species coincides with the period weaned pups are clustered on beaches at breeding sites for in spring before going to sea. Unlike in 2010, where both IAV detections were in adult female NES, half of the IAV RNA-positive animals in 2019 were pups and were both male and

female. Consistent with the 2010 report, none of the animals in 2019 showed signs of recognized IAV-like disease, suggesting that infection is asymptomatic or self-limiting. A caveat of our study is that our surveillance approach employs opportunistic sampling of stranded marine mammals, which may not represent IAV circulation patterns, timing, or disease manifestations in the entire population. The species and age bias may reflect sampling bias where in spring TMMC rescues mostly young NES and PHS pups that have been separated from their mothers, while in summer rescued animals are mostly one year old CSL showing signs of malnutrition or leptospirosis, cancer, domoic acid toxicity, or protozoal infections. Admission rates vary for each species and TMMC does not re-capture released animals. Seals that die on the beaches at breeding colonies in California are not examined to ascertain the cause of death, so influenza-related mortalities similar to those reported in harbor seals in New England and in the North Sea [1, 2, 21], may go undetected.

IAV antibody detection in three pinniped species in 2018 and 2019 suggests that IAV infections occurred in this period or that antibody titers from infections prior to 2018 are durable. The higher rate of antibody detections in NES, exceeding 40% in both years, compared to lower rates in CSL or PHS, suggests that NES are the most frequently infected of the three species, similar to previous observations [6, 22]. Frequent detection of IAV antibody in pups of all three species suggests either prior IAV exposure or transfer of maternal antibody from IAV-infected mothers [22]. The longevity of IAV maternal antibody in young marine mammals is not known. Three of the IAV RNA-positive animals seroconverted in the interval between blood collected at admission and collection of the second sample, suggestive of recent infection. Detection of IAV NP antibody in serum at intake from four out of 10 IAV RNA positive pups suggests that the magnitude or composition of IAV antibody was not sufficient to prevent IAV infection, resulting in viral RNA replication, or, alternately, that sampling occurred during infection before IAV clearance but after the production of IAV NP antibody. Our previous explant studies show that respiratory tract tissues from recently euthanized marine mammals, including NES, support IAV infection with multiple subtypes, including H1N1 [23].

The closest genetic identity of the 2019 NES IAV genome with contemporary pandemic human H1N1 shows that NES can be infected with pandemic human H1N1 viruses. The distant relatedness of the 2019 NES IAV genome with the two 2010 NES IAV genomes does not support the maintenance of a NES-exclusive IAV lineage. In contrast, the estimated time of most recent common ancestor, November, 2018, together with the phylogenetic analyses, suggested that the spillover event likely happened during the 2018/19 human influenza season since the NES virus closely matches the contemporaneously circulating human A/(H1N1) pdm09 6B.1A.1 subclade. This clade derived from the main clade 6B.1, or the A/Michigan/45/ 2015 vaccine virus relative clade. Since spring 2019, several genetic subclades within the 6B.1 clade were defined by specific amino acid substitutions in the HA and a new clade designated as 6B.1A became dominant. The 6B.1A clade contributed to a peak of influenza cases in humans in the 2018/19 season, where A/(H1N1)pdm09 represented 28% of cases [24]. After emergence of clade 6B.1A, many additional virus subclades that emerged encode a range of the HA amino acid substitutions, which were then assigned to 6B.1A1 - 6B.1A7 [16]. While the spillover event and intermediate infection chains are not discernable, the finding that A/ Northern elephant seal/California/ES4506NS/2019(H1N1) genetically matches the 2019 human 6B.1A.1 subclade in North America suggests that the spillover event occurred in the same influenza season, consistent with the tMRCA analysis. Subclade 6B.1A.1 viruses are characterized by S183P in HA, a mutation that occurs via mouse adaptation of A/(H1N1)pdm09 viruses, and may enhance receptor binding to bronchial α-2,3 sialic acid (SA)-linked receptors at the expense of decreased binding to α2,6 SA-linked receptors [25]. Whether this mutation

contributed mechanistically to the observed spillover and infection of NES with 6B.1A.1 A/(H1N1)pdm09 in the 2018/19 human influenza season is not known.

The mechanism(s) by which IAV is transmitted from humans into pinnipeds is also unclear. Generally, there are three main routes of influenza virus transmission: airborne, large droplets, and contact (direct and indirect contact) [26]. Notably, influenza A(H1N1)pdm09 virus has been detected in stool samples with positive viral culture from hospitalized human patients, suggesting viable shedding of the virus through feces [27]. In this study, exposure of the NES to human A(H1N1)pdm09 virus could have occurred through feces discharged from sewage-dumping ships, urban run-off, or on-shore exposure to another IAV reservoir also infected with human H1N1, including other seals or waterbirds. The tMRCA between the NES sequence and the closest human sequence (A/Pennsylvania/541/2018) was estimated at November 19, 2018 (ranging between October and December 2018) suggesting that the cross-species spillover likely happened during this timeframe.

In summary this study provides evidence of continued cross species transmission of pandemic influenza A virus from humans to free-ranging pinnipeds on the shores of California.

## Supporting information

**S1 Table. The reference set used for A(H1N1)pdm09 clade/subclade definitions.** (DOCX)

**S1 File. Maximum Likelihood phylogenetic trees of the PB2, PB1, PA, HA, NP, NA, M and NS genes of A(H1N1)pdm09 viruses sampled between 2016 and 2021 globally along with the reference sequences (orange) to clade/subclade designation.** The sequence in dark blue is the IAV genome from a Northern elephant seal in this study. Green sequences are IAV from other seals not part of this study. H1N1pdm09 subclade 6B.1A.1 represented by the reference 2019–2020 vaccine virus A/Brisbane/02/2018 is highlighted in yellow; the clade 6B.1A –in blue. Bootstrap supports are indicated next to the nodes, while branch lengths are scaled according to the number of nucleotide substitutions per site. (PDF)

**S2 File. Time-scale Bayesian MCC tree of the H1 gene of 165 A(H1N1)pdm09 isolates sampled globally between 2016–2021.** Bayesian posterior probability values > 70% are indicated. (PDF)

## Acknowledgments

We acknowledge Jesierose Poblacion, Carlos Rios, Christine Fontaine, Barbie Halaska, and the veterinary staff at TMMC, who conducted marine mammal sampling and provided swabs and serum under NOAA permit 18786–04. We thank Wendy Puryear, Nichola Hill, and Jonathan Runstadler at Tufts University, who provided HI protocols, and Randy Albrecht at the Icahn School of Medicine at Mount Sinai, who provided control sera for HI assays. Hon Ip at the National Wildlife Health Center provided the IAV H3N8 isolate. We are grateful to all authors from the originating laboratories responsible for obtaining the human specimens, as well as the submitting laboratories where the genome data were generated and shared via GISAID.

## Author Contributions

**Conceptualization:** Magdalena Plancarte, Lark L. Coffey.

**Data curation:** Magdalena Plancarte, Julie Baldassano, Lark L. Coffey.

**Formal analysis:** Magdalena Plancarte, Ganna Kovalenko, Julie Baldassano, Selina Carrillo, Ian Goodfellow, Eric Bortz, Lark L. Coffey.

**Funding acquisition:** Ian Goodfellow, Lark L. Coffey.

**Investigation:** Magdalena Plancarte, Ganna Kovalenko, Julie Baldassano, Jayeeta Dutta, Harm van Bakel.

**Methodology:** Magdalena Plancarte, Ganna Kovalenko, Lark L. Coffey.

**Project administration:** Lark L. Coffey.

**Resources:** Lark L. Coffey.

**Software:** Pádraig J. Duignan.

**Supervision:** Lark L. Coffey.

**Validation:** Magdalena Plancarte, Ganna Kovalenko, Selina Carrillo.

**Visualization:** Ana L. Ramírez, Ian Goodfellow, Lark L. Coffey.

**Writing – original draft:** Magdalena Plancarte, Lark L. Coffey.

**Writing – review & editing:** Magdalena Plancarte, Ganna Kovalenko, Ana L. Ramírez, Pádraig J. Duignan, Ian Goodfellow, Eric Bortz, Harm van Bakel, Lark L. Coffey.

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
