## [Decision Letter · Decision Letter 0]

10 Jan 2023

PONE-D-22-28455Human influenza A virus H1N1 in marine mammals in California, 2019PLOS ONE

Dear Dr. Coffey,

Thank you for submitting your manuscript to PLOS ONE. After careful consideration, we feel that it has merit but does not fully meet PLOS ONE’s publication criteria as it currently stands. Therefore, we invite you to submit a revised version of the manuscript that addresses the points raised during the review process.

We look forward to receiving your revised manuscript.

Kind regards,

Feng Wen

Academic Editor

PLOS ONE

Journal Requirements:

2. We note that Figure (2) in your submission contain [map/satellite] images which may be copyrighted. All PLOS content is published under the Creative Commons Attribution License (CC BY 4.0), which means that the manuscript, images, and Supporting Information files will be freely available online, and any third party is permitted to access, download, copy, distribute, and use these materials in any way, even commercially, with proper attribution. For these reasons, we cannot publish previously copyrighted maps or satellite images created using proprietary data, such as Google software (Google Maps, Street View, and Earth). For more information, see our copyright guidelines: http://journals.plos.org/plosone/s/licenses-and-copyright.

1. You may seek permission from the original copyright holder of Figure (2) to publish the content specifically under the CC BY 4.0 license.  

Natural Earth (public domain): " ext-link-type="uri" xlink:type="simple">http://www.naturalearthdata.com/"

Additional Editor Comments (if provided):

The manuscript is interesting. The authors must address the comments clearly.

Reviewers' comments:

Reviewer's Responses to Questions

**Comments to the Author**

1. Is the manuscript technically sound, and do the data support the conclusions?

Reviewer #1: Yes

2. Has the statistical analysis been performed appropriately and rigorously? 

Reviewer #1: Yes

3. Have the authors made all data underlying the findings in their manuscript fully available?

Reviewer #1: Yes

4. Is the manuscript presented in an intelligible fashion and written in standard English?

Reviewer #1: Yes

5. Review Comments to the Author

Reviewer #1: In this study, Plancarte et al. detected recent A/H1N1 virus from seals in North America. One sample could be sequenced and each of the eight gene segments is related to the human subclade 6B.1A.1 viruses. This study indicates important zoonotic spillover events from human to seals.

The manuscript is well-written, the experiment approaches in this study are sound but the methods for whole genome sequencing are too vague as this was referred to one citation. I think some important points could be clarified:

1. In the qPCR method, the CT value 45 was considered positive for influenza virus, how did this cut-off come about? Any value 40 is very likely negative. E.g. sample CSL-13161 in Table 1, both ELISA and HI are negative.

2. Fig 1, some samples showed positive HI titers for H3N8 and H5N2, such as ES-4302, ES-4348. These samples were also positive for H1. Could this be co-infection? Have the authors tried to sequence H3 and H5 by NGS?

3. In the phylogenetic analyses, concatenated eight segments were inferred using a Bayesian MCMC method. Influenza viruses carry segmented genome which undergo reassortment, also the rate of evolution varies among genes, it is incorrect to reconstruct a phylogeny based on whole genome. In fact, the authors have performed separate analyses based on each gene segment, but this is not stated in the Methods. The H1 (Suppl Fig S9) dated tree should be used in main figure, and not the concatenated genomes.

Minor comments:

1. Line 91, indicate what is the H1N1 strain.

2. Is the whole genome sequencing performed by NGS? Please add some details in the methods. And explain why Sanger sequencing of partial HA gene is needed then? Are the primers specific for H1?

3. Supp Fig. S9: Bayesian posterior probability is not expressed as %. Also, PP 0.95 is supported, so remove any values 0.95.

6. PLOS authors have the option to publish the peer review history of their article (what does this mean?). If published, this will include your full peer review and any attached files.

Reviewer #1: No

---

## [Author Response · Author response to Decision Letter 0]

28 Jan 2023

PONE-D-22-28455

Human influenza A virus H1N1 in marine mammals in California, 2019

PLOS ONE

Response to Reviewers

Thank you for considering our paper for publication in PLoS One. Please find below our response to comments in blue.

Please ensure that your manuscript meets PLOS ONE's style requirements, including those for file naming. The PLOS ONE style templates can be found athttps://journals.plos.org/plosone/s/file?id=wjVg/PLOSOne_formatting_sample_main_body.pdf and 

Response: Confirmed.

We note that Figure (2) in your submission contain [map/satellite] images which may be copyrighted. All PLOS content is published under the Creative Commons Attribution License (CC BY 4.0), which means that the manuscript, images, and Supporting Information files will be freely available online, and any third party is permitted to access, download, copy, distribute, and use these materials in any way, even commercially, with proper attribution. For these reasons, we cannot publish previously copyrighted maps or satellite images created using proprietary data, such as Google software (Google Maps, Street View, and Earth). For more information, see our copyright guidelines: http://journals.plos.org/plosone/s/licenses-and-copyright.

We require you to either (1) present written permission from the copyright holder to publish these figures specifically under the CC BY 4.0 license, or (2) remove the figures from your submission.

Response: We supplied a replacement figure (removed map) that complies with the CC BY 4.0 license.

Additional Editor Comments (if provided):

The manuscript is interesting. The authors must address the comments clearly.

Reviewers' comments:

Reviewer's Responses to Questions

Comments to the Author

1. Is the manuscript technically sound, and do the data support the conclusions?

Reviewer #1: Yes

2. Has the statistical analysis been performed appropriately and rigorously?

Reviewer #1: Yes

3. Have the authors made all data underlying the findings in their manuscript fully available?

Reviewer #1: Yes

4. Is the manuscript presented in an intelligible fashion and written in standard English?

Reviewer #1: Yes

5. Review Comments to the Author

Reviewer #1: In this study, Plancarte et al. detected recent A/H1N1 virus from seals in North America. One sample could be sequenced and each of the eight gene segments is related to the human subclade 6B.1A.1 viruses. This study indicates important zoonotic spillover events from human to seals.

The manuscript is well-written, the experiment approaches in this study are sound but the methods for whole genome sequencing are too vague as this was referred to one citation. I think some important points could be clarified:

1. In the qPCR method, the CT value 45 was considered positive for influenza virus, how did this cut-off come about? Any value 40 is very likely negative. E.g. sample CSL-13161 in Table 1, both ELISA and HI are negative.

RESPONSE: We agree with the reviewer that Cts of greater than 40 are likely negative. However, using a Ct of 45 as the upper limit is the standard for the NIH-CEIRS protocol we used. There was only 1 animal in our study that had a detectable Ct above 40 (the one the reviewer points out, animal CSL-13161). This animal stranded in 2016. We provided the data prior to 2019 mainly to show that IAV detections in marine mammals during that period were rare; as such this animal does not feature centrally into the story the paper describes. Of note, the animals in 2019 all had lower Cts below 40 and many were in the 20s, likely representing true IAV infections. 

2. Fig 1, some samples showed positive HI titers for H3N8 and H5N2, such as ES-4302, ES-4348. These samples were also positive for H1. Could this be co-infection? Have the authors tried to sequence H3 and H5 by NGS?

RESPONSE: Although we acknowledge co-infections are a possibility, we think the H3N8 and H5N2 titers more likely reflect antigenic cross reactivity in animals that experienced H1N1 infections than co-infections for the following reasons: 1) stochastically, co-infections with more than one IAV subtype are extremely rare, 2) H3N8 and H5N2 have not been reported previously in marine mammals, and 3) probably most importantly, the HA primers used for subtyping and the primers used to amplify the whole IAV genome by NGS were influenza-specific universal primers. This means that had other subtypes besides H1N1 been present, we would have detected them by both amplification approaches. We apologize for not including sufficient amplification and sequencing detail in the prior version for the reviewer to understand our approach. The manuscript has been updated to now include this information.

To methods, the following was added (in bold):

“Whole-genome sequencing was next performed using next generation sequencing from samples where HA sequencing was successful using methods described previously [14]. Briefly, RNA was extracted using the QIAamp Viral RNA Mini Kit (Qiagen, Valencia, CA). Following extraction, MS-RT-PCR amplification of influenza viral genome segments was performed with the Superscript III high-fidelity RT-PCR kit (Invitrogen, Carlsbad CA) according to manufacturer’s instructions using the Opti1 primer set: Opti1-F1 5’ GTTACGCGCCAGCAAAAGCAGG,Opti1-F2 5’GTTACGCGCCAGCGAAAGCAGG, Opti1-R1 5’GTTACGCGCCAGTAGAAACAAGG. These are influenza-specific universal primers that are complementary to the conserved 12–13 nucleotides at the end of all 8 genomic segments. The RT-PCR amplification parameters were: 2 min at 55°C, 60 min at 42°C, and 2 min at 94°C, followed by 5 cycles of (94°C/30 s; 44°C/30 s; 68°C/3.5 min), 26 cycles of (94°C/30 s; 57°C/30 s; 68°C/3.5 min), and a final extension for 10 min at 68°C. Amplicons were purified with 0.45X volume of Agencourt AMPure XP beads (Beckman Coulter, Brea, CA) and visualized on the Agilent Bioanalyzer or a 2% agarose gel. The concentration of purified amplicons was measured using the Qubit High Sensitivity dsDNA kit. After samples were normalized to a concentration of 0.2 ng/µl, adapters were added by tagmentation using the Nextera XT DNA library preparation kit (Illumina). Samples were purified using 0.7X of Agencourt AMPure XP Magnetic Beads and fragment size distributions were analyzed on a Bioanalyzer using the High Sensitivity DNA kit (Agilent). After bead-based normalization (Illumina, Hayward, CA) according to the manufacturer's protocols, sequence-ready libraries were sequenced in a paired-end run using the MiSeq v2, 300cycle reagent kit (Illumina). Complete genomes were assembled from next generation Illumina reads using a custom influenza genome assembly pipeline [14]. The genome assembly code is available at https://bitbucket.org/bakellab/flugap.

3. In the phylogenetic analyses, concatenated eight segments were inferred using a Bayesian MCMC method. Influenza viruses carry segmented genome which undergo reassortment, also the rate of evolution varies among genes, it is incorrect to reconstruct a phylogeny based on whole genome. In fact, the authors have performed separate analyses based on each gene segment, but this is not stated in the Methods. The H1 (Suppl Fig S9) dated tree should be used in main figure, and not the concatenated genomes.

RESPONSE: We thank the reviewer for this important comment. 

We agree that the concatenation approach should be used with caution depending on the aim of analyses and existing molecular datasets. In this case, we started the phylogenetic analysis with a maximum likelihood (ML) phylogenetic analysis of each gene segment. Comparing the tree topologies for each segment showed that the IAV genome detected in the Northern Elephant Seal in this study is not a reassortant virus and represents a pandemic A/H1N1 sequence. This is because the phylogenetic topologies of all eight segments were highly similar (homogeneous). Phylogenetic-based methods assume the absence of reassortment if sequences of different segments from the same isolate (virus) occupy similar positions in their respective phylogenies (congruence). If the sequences occupy conflicting phylogenetic positions (incongruence), it indicates their different origins. Given that these initial analyses showed no evidence of genomic reassortment, all segments were then concatenated and whole genomes were analyzed with Bayesian phylogenetic inference. The goal of this approach was to maximize phylogenetic resolution in our dataset by using viral genome information from all 8 segments instead of just H.

Although we still choose to present the concatenated tree in Fig 2 in the revised manuscript for the reasons highlighted above, we amended the Methods and Results sections to describe our approach more clearly and to explain that concatenation here is justified given we observed no evidence of reassortment when each segment was analyzed individually, which was our initial approach.

The following text was added (new in bold)

Methods:

“Given that our ML trees all showed that the IAV genome detected in the NES in this study is not a reassortant virus and belongs to A/H1N1 pandemic influenza type, we performed concatenation of all eight segments. The evolutionary relationships and timescale of the concatenated eight segments were inferred using a Bayesian Markov chain Monte Carlo (MCMC) method, as implemented in the BEAST v1.10.4 package [19].”

“The tree was visualized and annotated using FigTree v1.4.4 (http://tree.bio.ed.ac.uk/software/figtree). Additionally, to assess the reliability of the phylogenetic inference of the concatenated segments, an additional Bayesian tree was estimated for the H1 segment separately with the same dataset that was used for the concatenation approach and with the same model parameters.”

Results:

“Separate analyses of each of the eight segments (PB2, PB1, PA, H1, NP, N1, M, and NS) of the H1N1 virus from the NES in this study was inferred using the H1 classified phylogeny as a reference and shows that all gene segments group within the same clade (Supplementary Figures 1-8). In particular, the H1 HA phylogeny maintains similar topological accuracy to that of the concatenated tree, where individual clades are supported by high Bayesian posterior probabilities (BPP 90%) (Supplementary Figure S9). No reassortant events were detected in the H1N1pdm09 NES virus in our analyses.”

Minor comments:

1. Line 91, indicate what is the H1N1 strain.

RESPONSE:

The sentence was modified to add this information:

“Positive control sera from a ferret that had been experimentally inoculated with a pandemic H1N1 strain (A/California/04/2009) was provided by Dr. Randy Albrecht, Icahn School of Medicine at Mount Sinai, New York.”

2. Is the whole genome sequencing performed by NGS? Please add some details in the methods. And explain why Sanger sequencing of partial HA gene is needed then? Are the primers specific for H1?

RESPONSE:

Yes, whole genome sequencing was performed by NGS. 

We added detail to the methods explain the NGS approach. Please see response above.

To explain why we performed partial HA sequencing first, we modified this sentence:

We amplified and sequenced a portion of the IAV HA gene from RRT-PCR positive samples before conducting whole-genome sequencing that requires more time and effort.

The H typing primers should identify all H subtypes.

“For PCR, the forward primer 5’ GGRATGRTHGAYGGNTGGTAYGG 3’ was modified from a validated primer [12] to include HA sequences detected in California. The HARK reverse primer 5’ ATATGGCGCCGTATTAGTAGAAACAAGGGTGTTTT 3’ was first reported in Bragstad et al. [13] . These primers are designed to detect all HA subtypes.”

3. Supp Fig. S9: Bayesian posterior probability is not expressed as %. Also, PP 0.95 is supported, so remove any values 0.95.

RESPONSE: We feel that both percents and proportions are standard for expressing probability in visualization of phylogenies; as such, we choose to represent probabilities in Figure S9 as a proportion. For full transparency, we also feel it is best to report values that are lower than 0.95; as such, we also report values below 0.95 but above 0.70. As such, we have not modified Fig S9.

---

## [Editor Report · Decision Letter 1]

1 Mar 2023

Human influenza A virus H1N1 in marine mammals in California, 2019

PONE-D-22-28455R1

Dear Dr. Coffey,

We’re pleased to inform you that your manuscript has been judged scientifically suitable for publication and will be formally accepted for publication once it meets all outstanding technical requirements.

Kind regards,

Feng Wen

Academic Editor

PLOS ONE
---

## [Editor Report · Acceptance letter]

21 Mar 2023

PONE-D-22-28455R1 

Human influenza A virus H1N1 in marine mammals in California, 2019 

Dear Dr. Coffey:

I'm pleased to inform you that your manuscript has been deemed suitable for publication in PLOS ONE. Congratulations! Your manuscript is now with our production department. 

Kind regards, 

on behalf of

Dr. Feng Wen 

Academic Editor

PLOS ONE